# The Effect of Tropical Temperatures on the Quality of RNA Extracted from Stabilized Whole-Blood Samples

**DOI:** 10.3390/ijms231810609

**Published:** 2022-09-13

**Authors:** Yomani D. Sarathkumara, Daniel J. Browne, Ashton M. Kelly, David J. Pattinson, Catherine M. Rush, Jeffrey Warner, Carla Proietti, Denise L. Doolan

**Affiliations:** 1Centre for Molecular Therapeutics, Australian Institute of Tropical Health of Medicine, James Cook University, Cairns, QLD 4878, Australia; 2College of Public Health, Medical and Veterinary Sciences, James Cook University, Townsville, QLD 4811, Australia

**Keywords:** PAXgene^®^, Tempus^™^, blood RNA extraction, RT-qPCR, tropical, climate

## Abstract

Whole-blood-derived transcriptional profiling is widely used in biomarker discovery, immunological research, and therapeutic development. Traditional molecular and high-throughput transcriptomic platforms, including molecular assays with quantitative PCR (qPCR) and RNA-sequencing (RNA-seq), are dependent upon high-quality and intact RNA. However, collecting high-quality RNA from field studies in remote tropical locations can be challenging due to resource restrictions and logistics of post-collection processing. The current study tested the relative performance of the two most widely used whole-blood RNA collection systems, PAXgene^®^ and Tempus™, in optimal laboratory conditions as well as suboptimal conditions in tropical field sites, including the effects of extended storage times and high storage temperatures. We found that Tempus™ tubes maintained a slightly higher RNA quantity and integrity relative to PAXgene^®^ tubes at suboptimal tropical conditions. Both PAXgene^®^ and Tempus™ tubes gave similar RNA purity (A260/A280). Additionally, Tempus^™^ tubes preferentially maintained the stability of mRNA transcripts for two reference genes tested, *Succinate dehydrogenase complex, subunit A* (*SDHA*) and *TATA-box-binding protein* (*TBP*), even when RNA quality decreased with storage length and temperature. Both tube types preserved the rRNA transcript *18S ribosomal RNA* (*18S*) equally. Our results suggest that Tempus^™^ blood RNA collection tubes are preferable to PAXgene^®^ for whole-blood collection in suboptimal tropical conditions for RNA-based studies in resource-limited settings.

## 1. Introduction

Gene expression profiles from whole-blood-derived RNA have proven useful as a molecular signature reflecting physiological and pathological changes in the body [1]. Immune signatures for various diseases and metabolic states, including biomarkers of disease, disease prognosis, or therapeutic efficacy, have been identified by blood transcriptional profiling [2,3,4]. Since blood transcript profiles are reproducible, cost-effective, and easy to implement, they can be rapidly translated into clinical practice [5]. High-quality and intact RNA is imperative for both traditional molecular diagnostics and high-throughput transcriptome sequencing techniques, such as quantitative PCR (qPCR) [6] and RNA-sequencing (RNA-seq) [7]. High-quality RNA can be readily obtained from fresh blood when processed immediately following collection. However, RNA quality can be adversely impacted by processing delays and complex logistical issues. Remote tropical locations are particularly challenging due to locality and resource restrictions [8]. Pre-extraction factors that influence gene expression or lead to RNA degradation include processing delays and storage conditions (i.e., duration and temperature) [9,10,11,12] since blood RNA is highly susceptible to enzymatic degradation and oxidative damage [13,14].

Commercially available blood collection systems with additives that stabilize RNA have been developed to address RNA stability, resulting in significantly enhanced quantity and quality of RNA extracted from whole blood [15,16]. The two most widely used blood RNA stabilizing systems are PAXgene^®^ and Tempus^™^ [17,18]. These systems contain proprietary solutions that lyse cells, inactivate RNases, and minimize changes in gene expression [19]. Those systems are designed to facilitate long-term whole-blood storage at low temperatures (−80 °C), removing the necessity of isolating RNA immediately post-collection and allowing batched processing [8,20]. However, each system has different stabilization efficiencies, thus impacting the resultant transcriptional profiles [21]. Several studies have investigated the relative performance of PAXgene^®^ and Tempus^™^ in specific suboptimal conditions [17,20]. There is, however, an unmet need to identify the optimal RNA stabilization system for use in suboptimal tropical conditions, including high temperatures and extended times before storage at −80 °C.

In this study, we compared the effects of extended storage times and high temperatures simulating suboptimal tropical conditions before freezing of whole-blood RNA stabilized in either the PAXgene^®^ or Tempus^™^ systems.

## 2. Results

### 2.1. Magnetic-Bead and Spin-Column-Based RNA Purification Systems Extracted Equivalent Whole-Blood RNA Quantity, Quality, and Purity

The quantity, quality, and purity of RNA extracted were evaluated using two common RNA isolation systems [18]: spin-column-based (PAXgene^®^ QIAGEN/BD; Tempus^™^ Applied Biosystems) or magnetic-bead-based (MagMAX^™^ Life Technologies; compatible with either PAXgene^®^ or Tempus^™^ collection tube) systems. RNA concentration (ng/μL) and purity (A260/A280 and A260/A230) were evaluated by spectroscopic quantification using NanoPhotometer^®^ N60 (Implen, München, Germany). RNA integrity number (RIN) was determined using an Agilent 2100 Bioanalyzer (Agilent Technologies). Total RNA yield (ng) was normalized to whole-blood volume collected in each blood RNA collection system (i.e., 2.5 mL in PAXgene^®^ vs. 3 mL in Tempus^™^). All RNA samples were extracted fresh post-collection for the comparison of RNA isolation protocols.

We found the RNA isolation protocol did not affect the amount of total RNA extracted, the RNA quality, or the purity of RNA (normalized total RNA: *p* = 0.875, RIN: *p* = 0.124, A260/A280: *p* = 0.101, A260/A230: *p* = 0.318, MagMAX^™^ vs. columns, unpaired *t*-test) (Appendix A). Additionally, all samples had RIN of >7.0 and an A260/A280 ratio between 1.98–2.15, suggesting recovery of high-quality and high-purity RNA from both RNA isolation systems. A260/230 ratios were used as a secondary measurement detecting presence of residual phenol/ethanol, salts, and carbohydrates that can affect RNA quality. We observed differences between column vs. MagMAX^™^ extraction methods although differences were not significant. Comparatively, better values were obtained with Tempus^™^ column extractions. However, RNA yields obtained from the Tempus^™^ tubes were significantly lower than those obtained with PAXgene^®^ tubes (*p* = 0.008, unpaired *t*-test), suggesting that the PAXgene^®^ tubes gave higher RNA yields than Tempus^™^ when extracted in optimal laboratory conditions (Appendix A). Given the high quality of the extracted RNA and the advantages of extracting in a 96-well-plate format [6], magnetic bead-based (MagMAX^™^) RNA purification system compatible with PAXgene^®^ and Tempus^™^ systems were used for the rest of the extractions.

### 2.2. A Higher Quantity of RNA Was Obtained Using Tempus^™^ Blood RNA Tubes in Suboptimal Tropical Conditions

Next, we compared the quantity, quality, and purity of RNA extracted from whole-blood samples stored in suboptimal tropical conditions collected in PAXgene^®^ and Tempus^™^ tubes. To simulate suboptimal tropical conditions, matched whole-blood samples were collected in either PAXgene^®^ or Tempus^™^ tubes and stored at different temperatures (25, 30, 35, or 40 °C) for different lengths of time (0, 1, 5, 7, or 10 days) before final storage at −80 °C for later extraction. These samples were compared to matched samples immediately frozen at −80 °C for later extraction (D1/Control) or unmatched samples collected and processed post collection immediately in optimal laboratory conditions (D0/Fresh). The tube type had no effect on RNA quantity (i.e., normalized RNA concentration) when extracted in Fresh (*p* = 0.065, paired t-test) or Control (*p* = 0.274, paired *t*-test) conditions (Figure 1A). However, we found that tube type significantly affected normalized RNA concentration (*p* < 0.0001; Table 1, Model 1) in samples subjected to suboptimal tropical conditions.

To investigate these data further, we applied three multiple linear regression models to evaluate the effects of explanatory variables (i.e., tube type, storage times, temperature, and biological subject) on normalized RNA concentrations, A260/A280 ratios, and RIN values (Table 1). Model 1 explained 76% of the variation in extracted RNA quantity (*R*^2^ = 0.764, *p* < 2.2 × 10^−16^). Tempus^™^ tubes had a significant effect on the RNA yield (*p* = 1.15 × 10^−6^), and temperature variation in tube type significantly impacted RNA yield (*p* = 0.0003; Table 1, Model 1). Additionally, we applied separate linear models to each temperature condition to explore the effect of the temperature on the tube type (Appendix A). We found that the concentration of RNA extracted from whole-blood collected in Tempus^™^ tubes was significantly greater than for PAXgene^®^ tubes at all evaluated temperatures (*p* = 7.16 × 10^−6^ (25 °C); *p* = 0.0004 (30 °C); *p* = 0.033 (35 °C); *p* = 0.023 (40 °C), Appendix A). As a secondary measurement of RNA concentration, we obtained the RNA concentration readings from Agilent Bioanalyzer. Similarly, Tempus^™^ tubes gave higher RNA yields when measured with Agilent Bioanalyzer (*p* = 0.001; Appendix A). Interestingly, RNA concentration measurements by spectrophotometer and bioanalyzer were more strongly correlated in Tempus^™^ than in PAXgene^®^ tubes (*p* = 0.0002, *R^2^* = 0.270 (PAXgene^®^); *p* = 9.3 × 10^−9^, *R^2^* = 0.518 (Tempus^™^)); Figure 1B), suggesting that tube-specific contents influence concentration measurements.

When considering RNA purity, all extracted RNA samples had A260/A280 ratios >2 regardless of tube type, indicating a high purity under all test conditions (Figure 1A, middle panel). There was no difference in A260/A280 ratios between tube type for RNA extracted from fresh and control RNA samples ((*p* = 0.480, (Fresh); *p* = 0.111, (Control), paired *t*-test). However, at tropical storage and temperature conditions, the tube type (*p* = 0.011) and the storage temperature on tube type (*p* = 0.003) had significant effects on A260/A280 ratios as per the multiple linear regression model (Table 1, Model 2). We used A260/230 ratio as a secondary measurement of RNA purity although with low quantities, as ng/µL RNA A260/230 ratios are highly variable A260/230 ratios (Appendix A). A260/230 ratios are irrelevant for RNA quality and only describe dissolved solvent concentration relative to RNA. Linear models further indicated that the higher temperatures decreased purity as evidenced by A260/A280 ratio (30 °C (*p* = 0.007), 35 °C (*p* = 0.011) and 40 °C (*p* = 4.89 × 10^−5^), Appendix A). Taken together, these data demonstrated that higher RNA yields are extracted from Tempus^™^ blood tubes compared to PAXgene^®^ tubes in suboptimal tropical conditions. These data also showed that RNA yields significantly decreased with increasing temperature in both PAXgene^®^ and Tempus^™^ tubes.

### 2.3. A Higher Quality of RNA Was Obtained Using PAXgene^®^ Tubes in Optimal Laboratory Conditions

We determined if high RNA quality was preserved using PAXgene^®^ or Tempus^™^ tubes in suboptimal tropical conditions and compared the RNA quality with samples extracted in optimal laboratory conditions. RIN values declined over time and temperature irrespective of the whole-blood collection tube (Figure 1A). PAXgene^®^ had significantly higher RIN values in fresh (*p* = 0.013, paired *t*-test) and control conditions (*p* = 0.001, paired *t*-test) compared to Tempus^™^. The electropherograms showed comparable results for different conditions applied on PAXgene^®^ and Tempus^™^ systems (Figure 1C). Ribosomal RNA bands were clearly visible in fresh extractions and control samples. Most RNA eluates stored at room temperature (25 °C) for 5–7 days obtained RIN values around 5–6 with visible *18S* and *28S* bands. In contrast, RNA stored at 40 °C did not show distinct rRNA banding. These data demonstrated that higher-quality RNA was obtained with PAXgene^®^ tubes (compared to Tempus^™^) when RNA was extracted post collection immediately or when samples were maintained at the optimal storage conditions as recommended by the manufacturers.

### 2.4. The Highest Quality RNA Was Obtained Using Tempus^™^ Blood RNA Tubes in Suboptimal Tropical Conditions

A multiple linear regression model was built to explore the effects of suboptimal tropical conditions on RNA integrity to determine the effect of tube type, storage temperature, and storage time on RNA integrity (measured by RIN values) (Table 1, Model 3). Model 3 explained approximately 79.7% (*R*^2^ = 0.797) of the variation in RIN values. RIN values decreased significantly over time (*p* = 0.045), decreasing by 0.862 per day (Model 3, days estimate = −0.148; e^−0.148^ = 0.862). RIN values also decreased significantly with the increasing of temperature (*p* < 0.0001), decreasing by 0.921 (Model 3, temperature estimate = −0.082; e^−0.082^ = 0.921) for each degree Celsius (°C) increase in temperature. These data demonstrated that RNA integrity significantly decreased with the length of storage time and temperature.

The impact of storage time on RNA integrity (RIN values) was also analysed at each temperature for each tube type (Appendix A). RIN values were significantly higher in Tempus^™^ than PAXgene^®^ tubes at 30 °C (*p* = 0.0001) and 35 °C (*p* = 0.001). However, no significant differences in RIN at 25 °C (*p* = 0.787) or 40 °C (*p* = 0.399) was found for tube types (Figure 1A). In agreement with previously published studies [9,16], these results demonstrated that RNA integrity is temperature-sensitive, and that both tube types produced low-quality RNA at increased storage times and temperatures. Nevertheless, our data suggest that Tempus^™^ tubes may provide better RNA integrity (higher RIN values) under certain suboptimal tropical conditions compared to PAXgene^®^ tubes.

### 2.5. Tempus^™^ Tubes Maintain mRNA Integrity across Suboptimal Tropical Conditions

In order to validate our RNA quality measurements, we quantified mRNA and rRNA extracted from PAXgene^®^ and Tempus^™^ tubes using RT-qPCR. We tested the relative mRNA abundance of two human reference genes, *Succinate dehydrogenase complex, subunit* A (*SDHA*) and *TATA-box-binding protein* (*TBP*), and one rRNA transcript, *18S ribosomal RNA* (*18S*). The RNA concentration of all samples was normalized at pre-cDNA synthesis (i.e., at 30 ng/μL). Hence, an increasing cycle threshold (Ct) value indicated a decreasing relative transcript quality rather than abundance [22]. We tested RT-qPCR primer sets designed to amplify different-sized fragments of the same target gene (i.e., amplicons between 100–300 bp) and differences in the relative RNA quality (e.g., increased Ct values) would be expected to be intensified when assaying genes with primers amplifying larger amplicons.

There was no significant difference in the mean Ct values between the tube types for Control samples from smaller amplicons (*p = ns,* paired *t*-test: D1/Control, 100–200 bp) or with larger amplicons (*p = ns*, paired *t*-test: D1/Control, 200–300 bp, Figure 2). For example, mean Ct values obtained for larger amplicons (200–300 bp) tested in PAXgene^®^ tubes (22.32 (*18S*), 25.48 (*SDHA*), 27.39 (*TBP*)) and in Tempus^™^ tubes (21.33 (*18S*), 25.59 (*SDHA*), 27.59 (*TBP*)) were largely consistent. In addition, we found no statistically significant difference between matched fresh and control samples between PAXgene^®^ and Tempus^™^ tubes (Appendix A). However, the Ct value varied significantly at tropical storage conditions across all three tested genes across short (100–200 bp) and medium amplicons (200–300 bp) (Figure 2). Tempus^™^ tubes maintained significantly higher transcript stability, as indicated by lower Ct values obtained for three tested genes compared to PAXgene^®^ tubes at suboptimal tropical conditions (Appendix A).

Multiple comparison testing found that the *18S* rRNA Ct values were not statistically significantly influenced by incubation temperature or duration when rRNA was collected in either PAXgene^®^ or Tempus^™^ tubes. In contrast, mRNA (*SDHA* and *TBP*) collected in PAXgene^®^ tubes were significantly impacted by storage time and temperature compared to Tempus^™^ tubes (Appendix A).

We showed that the RNA degraded samples, as indicated by the decreasing level of RIN, had higher Ct values (Figure 3A). The Ct shifted towards higher cycle numbers for *SDHA* with larger amplicons than short- and medium-length amplicons, which was much more evident in PAXgene^®^ tubes than in Tempus^™^ (Figure 3A). These results indicated that relative overall stability in terms of mRNA expression levels was maintained in Tempus^™^ compared to PAXgene^®^ tubes. A similar relationship between RIN and Ct values was observed for *TBP* (Appendix A). However, as clearly shown in Appendix A, both tube types had Ct < 30 for all product lengths for *18S,* suggesting both PAXgene^®^ and Tempus^™^ tubes preserved rRNA at suboptimal tropical conditions. As indicative of decreasing relative transcript quality, increasing Ct values were validated by correlating change in Ct values with RIN. We considered the change in Ct values (∆Ct) as the difference between samples collected under suboptimal tropical conditions and the mean of the control samples. Strong statistically significant correlations were found between ∆Ct and RIN for all tested genes (Figure 3B). These negative correlations indicated that with the decreasing RIN values, the ∆Ct of 200–300bp amplicons increased, thus validating the use of RT-qPCR to assess the quality of the RNA. Taken together, these data demonstrate that Tempus^™^ collection tubes better maintain mRNA stability in suboptimal tropical conditions even despite a decreasing RIN.

To test if the presence of PCR inhibitors, which are often co-extracted from whole blood (e.g., haemoglobin, lactoferrin, anticoagulants, etc.) [23,24], could have contributed to these results, RT-qPCR was performed on a log_2_ serial dilution of undiluted extraction eluent. We considered that a trendline gradient of Ct values relative to the dilution greater than −3.3 (i.e., E’ < 100%) was indicative of the presence of PCR inhibitors [23]. There was no apparent effect of inhibitors in both PAXgene^®^ or Tempus^™^ tubes when the samples were diluted below 60 ng/uL (Appendix A). These data demonstrated that our findings were unlikely to be a consequence of inhibitors present in the RT-qPCR reaction.

In summary, our data showed that Tempus^™^ tubes maintained a higher RNA quantity and integrity comparatively to PAXgene^®^ tubes when RNA is stored in suboptimal tropical conditions. Furthermore, Tempus^™^ tubes maintained stability of mRNA in conditions where RNA samples were heavily degraded as indicated by RIN. Taken together, this study establishes that the Tempus™ blood RNA collection system resulted in a better quality of RNA and enhanced stability of mRNA when whole-blood samples are stored under suboptimal tropical conditions.

## 3. Discussion

Gene expression profiling with molecular techniques such as RT-qPCR and next-generation sequencing requires high-quality intact RNA. It is well-established that the pre-analytical variables in blood sample collection and processing have profound effects on RNA quality that may consequently introduce substantial technical bias for molecular analysis [11,25]. Pre-analytical handling of blood samples and storage can be challenging in tropical remote field study settings where freezing at −80 °C immediately post-collection may not be an option. Here, we evaluated PAXgene^®^ and Tempus^™^ blood RNA stabilization tubes for preserving RNA quantity, purity, quality, and gene transcript stability at suboptimal tropical conditions.

According to the respective manufacturers, PAXgene^®^ blood RNA tubes effectively stabilize RNA for up to three days at room temperature, five days at 2–8 °C, and up to 11 years at −20 °C or −70 °C, whilst Tempus™ blood RNA tubes stabilize RNA for up to five days at room temperature and at least a week at 4 °C or −80 °C for long-term storage [26,27]. Duale et al. showed that the RNA yield, quality, and integrity were stable up to six years of storage at −80 °C in Tempus™ blood RNA tubes [8].

This study determined the impact of warm tropical temperatures (25, 30, 35, and 40 °C) and prolonged storage times (0, 5, 7, and 10 days) on total RNA yield, purity, quality, and transcript stability of the two most widely used commercially available blood RNA stabilizing systems, PAXgene^®^ and Tempus^™^. These conditions were selected to simulate conditions in field sites in tropical or subtropical regions for a 10-day field trip, representing a challenging situation for preserving RNA. The performances of commercially available kits with columns (spin-column-based) and MagMAX^™^ (magnetic-beads-based) protocols were used to extract RNA from blood collected in PAXgene^®^ and Tempus^™^ tubes. The total RNA yield, RNA integrity (RIN), and purity were used as performance measures.

In experiment A, both columns vs. MagMAX^™^ extraction methods had no significant differences in normalized total RNA yields, RIN, and A260/280 ratios in PAXgene^®^ and Tempus™ tubes. However, total RNA obtained from Tempus™ tubes was significantly lower than PAXgene^®^ tubes. We observed comparable average OD 260/280 ratios and RIN > 7 for column and MagMAX^™^ extraction methods for blood collected in PAXgene^®^ and Tempus™ tubes. One limitation of experiment A was that only two biological subjects were evaluated for Tempus^™^ column vs. MagMAX^™^ extractions due to the unavailability of Tempus™ blood collection tubes of the same batch. An A260/A280 ratio between 1.8 to 2.2 indicates highly purified RNA with minimum DNA contamination [28]. These data demonstrated that a similar quantity, quality, and purity of RNA could be obtained using either spin-column or magnetic-bead RNA purifications when purifying RNA from whole-blood collected in either PAXgene^®^ or Tempus^™^ systems. Higher RIN value indicates better RNA integrity, and RIN values above seven are considered ideal for high-throughput downstream applications [29,30]. However, RNA samples with a RIN of five have been used in gene expression studies [14,31]. MagMAX^™^ RNA extractions were used as the method of RNA purification for the rest of the study.

Both PAXgene^®^ and Tempus^™^ tubes gave similar RNA yields and purity (A260/A280) when the samples were extracted fresh or post freezing, whilst RIN values were significantly higher in blood samples extracted from PAXgene^®^ than Tempus^™^. The reason for this difference is not yet known. Overall RIN values were much lower in control samples compared to freshly extracted RNA. A previous study investigating the impact of storage duration (24, 32, and 40 h) and storage temperatures (24 °C, 4 °C, and −80 °C) of whole blood collected in heparin tubes on the qualities of DNA and RNA showed that RNA integrity declined dramatically when the samples were frozen [9]. Freezing blood samples will lead to irreversible cellular damages, causing osmotic and ice injuries of red blood cells due to water crystallization [32]. Activated intracellular enzymes such as RNases can be released upon thawing, resulting in RNA degradation [33].

The linear regression model on normalized RNA concentrations indicated that the Tempus^™^ tubes result in a higher RNA yield than PAXgene^®^, while we demonstrated that the tube type and temperature significantly affect RNA yields. However, similar RNA purities were obtained from both tube types. A similar study by Duale et al., comparing PAXgene^®^ vs. Tempus^™^ tubes stored for 0, 2, 5, and 7 days at RT (~22 °C) and then stored at −80 °C until extraction, showed that RNA yields collected in the Tempus^™^ tubes were consistently higher than PAXgene^®^ tubes. However, RNA quality (average 260/280 ratios and RIN values) was similar in both systems [17]. Consistent with our study, other studies have reported that higher RNA yields were obtained with Tempus^™^ tubes compared to PAXgene^®^ tubes [34,35,36,37]. However, we observed low A260/230 ratios (less than 2.0) for both tube types in our study. This ratio is decreased in the presence of residual phenol, salts, and carbohydrates that can affect the accuracy of downstream application and used as a secondary measurement for RNA purity [38]. Historically, low A260/A230 ratios are reportedly attributed to the high salt content of the elution buffers contained in PAXgene^®^ extraction kits [16,17,34,35] and as well as in Tempus^™^ extraction kits [16,22,39].

Our results indicated a gradual decrease of RNA quality in terms of RIN values over storage duration and increased temperatures in both tube types. However, RNA extracted from Tempus^™^ tubes had improved RINs compared to PAXgene^®^ at suboptimal tropical conditions. RNA integrity was influenced mainly by increased storage temperatures at higher temperatures. It has been well-documented that RNA molecules are sensitive to physical degradation due to high temperatures [40]. However, the effects of higher temperatures on Tempus^™^ and PAXgene^®^ blood RNA stabilizing systems have not been previously studied. Our data suggested that good quality RNA (average RIN >5) can be obtained in both tube types when samples are kept at 25 °C room temperature for up to 5 days of storage duration. Overall, we demonstrated that satisfactory amounts of good-quality RNA can be achieved using blood RNA stabilizing systems in warm tropical temperatures (25–30 °C) and at storage times up to a week. When the storage temperatures are above 30 °C, the RNA quality drops significantly and may not be adequate in downstream applications. RT-qPCR data further demonstrated that PAXgene^®^ tubes do not preserve mRNA with the same efficiency as Tempus^™^ tubes, but both tubes equally preserved rRNA from degradation in suboptimal tropical conditions.

In most low-resource settings, microscopy and serological assays such as ELISAs remain the standard methods for diagnosis of tropical infections, especially in low-income and middle-income countries, despite limited sensitivity and specificity. More sensitive molecular methods have potential to inform disease, diagnosis, and treatment and to facilitate field-based intervention and biobanking studies (i.e., large-scale field trials). This report provides important information to facilitate such studies. In particular, our data show that the Tempus^™^ blood RNA collection system resulted in higher-quality RNA and maintained more consistent stability of mRNA when whole-blood samples were stored under suboptimal tropical conditions as compared with the PAXgene^®^ system.

## 4. Materials and Methods

### 4.1. Sample Collection

Whole blood was collected from healthy adult volunteer donors into Tempus^™^ (3 mL) Blood RNA tubes (Applied Biosystems, Foster City, CA, USA) or PAXgene^®^ (2.5 mL) Blood RNA tubes (PreAnalytiX, QIAGEN/BD, Hombrechtikon, Switzerland) according to manufacturer’s instructions. Briefly, whole blood was collected directly into each tube by standard venepuncture and immediately shaken vigorously for 10 s to ensure that the stabilizing reagent made uniform contact with the sample as per the manufacturer’s instructions.

### 4.2. Experimental Design

Experiment A compared RNA yields, purity, and integrity from MagMAX™ extractions using spin columns or magnetic beads (Figure 4, top panel). All blood samples collected for experiment A were kept for two hours at 25 °C room temperature (RT) after collection and freshly extracted (i.e., no storage at −80 °C). Experiment B evaluated RNA yields, purity, and integrity in whole-blood samples stored at different temperatures (25, 30, 35, or 40 °C) and storage times (0, 5, 7, or 10 days; Figure 4). All samples collected for experiment B were frozen at −20 °C overnight then transferred to −80 °C until RNA extraction with magnetic-beads-based MagMAX^™^. All blood samples stored in PAXgene^®^ tubes were thawed for 2 h at room temperature, whereas Tempus^™^ tubes were thawed for 30 min on ice prior RNA isolation.

### 4.3. RNA Extraction

#### 4.3.1. Column-Based RNA Purification

Total RNA from whole blood collected in PAXgene^®^ tubes was extracted according to manufacturer’s instructions using PAXgene^®^ Blood RNA Kit (PreAnalytiX, QIAGEN/BD, Hombrechtikon, Switzerland), which included DNase I treatment. Total RNA was eluted in 40 μL elution buffer. According to the manufacturer’s instructions, total RNA from blood collected in Tempus^™^ tubes was extracted using the Tempus^™^ Spin RNA Isolation Kit (Applied Biosystems, CA, USA). RNA was eluted in 90 μL of elution solution. DNase treatment was an optional step in the Tempus^™^ column extraction system and therefore not included, as genomic DNA contamination using this procedure is minimal (less than 0.05% by weight) according to the manufacturer’s specifications.

#### 4.3.2. Magnetic-Bead-Based RNA Purification (MagMAX™)

Total RNA from whole blood was extracted from PAXgene^®^ Blood RNA Tubes using MagMAX^™^ for Stabilized Blood Tubes RNA Isolation Kit (Life Technologies, CA, USA) according to the manufacturer’s protocol, including a TURBO^™^ DNase and protease step. RNA was isolated using MagMAX^™^ for Stabilized Blood Tubes RNA Isolation Kit, compatible with Tempus^™^ Blood RNA tubes (Life Technologies, CA, USA) following the manufacturer’s protocol with TURBO^™^ DNase treatment. All extracted RNA samples were stored at −80 °C. The technical characteristics of each extraction method are summarised in Appendix A.

### 4.4. RNA Yield, Purity and Integrity

RNA concentration (ng/μL), A260/A280, and A260/230 ratios to indicate RNA purity were measured by spectroscopic quantification using NanoPhotometer^®^ N60 (Implen, München, Germany). RNA integrity was measured using the Agilent 2100 Bioanalyzer (Agilent Technologies) and the Eukaryote Total RNA Nano assay, complementing RNA 6000 NanoChip kit (Agilent Technologies, CA, USA), following the manufacturer’s instructions. The RNA integrity number (RIN) was calculated by the Agilent 2100 Expert software (Version B.02.10.SI764, Agilent). The RIN ranges from 1 to 10; an RIN of fully intact RNA is 10, and an RIN of completely degraded RNA is 1.

### 4.5. Reverse Transcription Quantitative PCR (RT-qPCR)

#### 4.5.1. Reverse Transcription

All reverse transcription (RT) reactions were conducted using the SuperScript IV^™^ First-Strand Synthesis System^™^ (ThermoFisher Scientific, Waltham, MA, USA). All samples were primed with 37.5 ng of random hexamers and 10 mM dNTPs at 65 °C for 5 min and then 4 °C for 1 min. Reverse transcription was then performed using the SuperScript IV^™^ reverse-transcriptase (SSIV) for 10 min at 23 °C, 10 min at 50 °C, and 10 min at 85 °C. SSIV concentration was assessed at 20 U (20 units/μL RNA) reactions compared with 5 U reactions (Appendix A) as previously described [41]. All subsequent RNA samples were reverse transcribed at 30 ng/μL (Appendix A) using 5 U reactions in 15 μL total volume reactions for test conditions. All cDNA samples were stored at 4 °C.

#### 4.5.2. Quantitative PCR (qPCR)

qPCR was run with 5 μL total reaction volume using SsoAdvanced SYBR^®^ SuperMix (Bio-Rad, Hercules, CA, USA), which facilitates excellent reaction efficiencies [6]. All reactions contained 0.5 μM of desalt-grade primers (Sigma-Aldrich) with 0.75 ng/uL sample cDNA. Each sample was run in technical triplicate replicate, followed by a melt curve analysis to ensure primer specificity. Primers used for the RT-qPCR assays were sourced from Primer Bank^™^ [42] (Appendix A). Primer efficiencies were calculated as per MIQE guidelines [43] as previously published from cDNA standards [6]. Reaction efficiency was calculated from log_2_ dilutions of pooled cDNA from 1 × 10^6^ unstimulated PBMCs. The PCR cycling program included an enzyme activation step at 95 °C for 2 min and then 40 cycles of annealing and extension at 95 °C for 15 s and 60 °C for 30 s, respectively. The cycle threshold (Ct) value was set to 0.3 ΔRN, and a pooled cDNA positive control was included across all plates to ensure reproducibility. qPCR was performed using the QuantStudio 5 Real-Time PCR system running QuantStudio Design and Analysis Software (v1.5.1, Applied Biosystems).

### 4.6. Statistical Analysis

Statistical analyses were performed using R statistical software (https://www.r-project.org/ (accessed on 24 November 2021, RStudio Inc., Boston, MA, USA, Version 1.4.1103). Unpaired *t*-tests were used to compare the data from spin-column-based and magnetic-bead-based MagMAX^™^ RNA extractions to determine any significant difference between RNA isolation systems for total RNA yield, A260/A280 ratios, and RNA integrity. Statistical significance was defined using *p*-values < 0.05. Normal distribution of data and normality of residuals were evaluated using the Shapiro–Wilks test.

The average Ct values for each replicates/triplicate and targets with Ct-values > 35 or undefined were considered beyond the limit of detection (LOD) and removed from the analysis [17]. Paired *t*-tests determined the differences of RNA yields normalized to input whole blood volume, A260/A280 ratios, RIN values, and Ct values for short and medium lengths of all housekeeping genes between the two RNA stabilization systems (PAXgene^®^ and Tempus^™^) for RNA extracted on fresh and control conditions. Whole-blood samples processed post collection immediately at optimal laboratory conditions without −80 °C storage are referred to as D0/Fresh. Samples frozen immediately at −80 °C for later extraction are referred to as D1/Control.

Multiple linear regression models were fit to investigate the overall relationship of independent variables at experimental conditions (i.e., tube type, temperature, and storage time) on normalized RNA yields, A260/A280 ratios, RIN values, or Ct values. To further explore the significant interaction between temperatures and tube type, separate analyses were performed at each temperature. To compare the changes of Ct values on control and test conditions on PAXgene^®^ and Tempus^™^ tube types, we performed a two-way analysis of variance with multiple comparisons.

## 5. Conclusions

We conclude that collection of whole blood samples in Tempus^™^ tubes is the preferred system of choice for gene expression and molecular studies in rural and remote resource-limited settings where electricity and storage facilities are compromised. Our findings are especially relevant to research on RNA biology, which could help future directions of diagnosis, treatment, and interventions against diseases that are prevalent in tropical countries, including neglected tropical diseases.

## Figures and Tables

**Figure 1 ijms-23-10609-f001:**
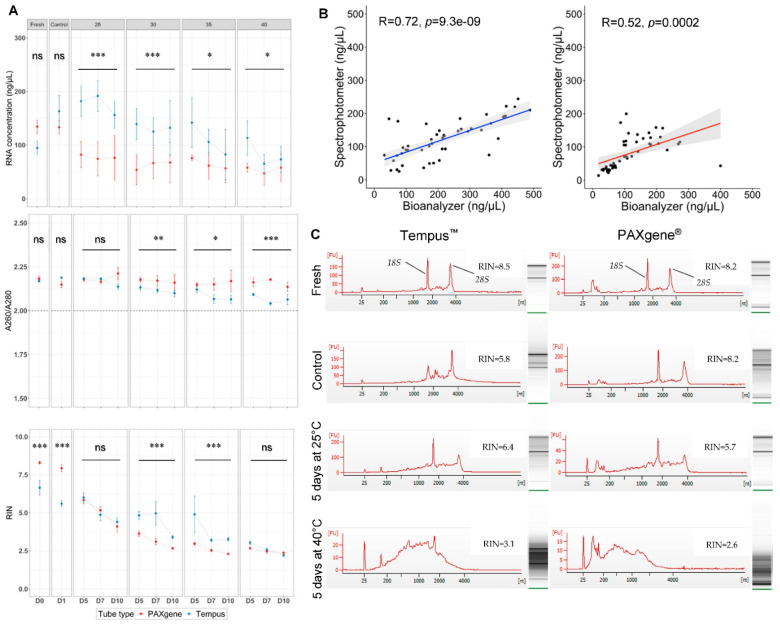
Comparison RNA concentration, purity, and integrity between PAXgene^®^ and Tempus^™^ blood collection systems under suboptimal tropical storage conditions. (**A**) Mean RNA yield normalized to whole-blood volume (ng/μL), A260/A280 ratios determined spectrophotometrically, and RNA integrity number (RIN) across the different conditions (Fresh (*n* = 8), Day1/Control; immediately frozen at −80 °C for later extraction and various temperatures (25, 30, 35, 40 °C) (matched subjects *n* = 3)). (**B**) Scatter plots revealed correlations between normalized RNA concentration (ng/uL) by Spectrophotometer or Bioanalyzer for PAXgene^®^ (left panel) and Tempus^™^ (right panel). Pearson’s correlation assessed correlations between variables. (**C**) Electropherograms showed two distinct peaks (*28S* and *18S*), gel images showed two bands comprising the *28S* and *18S* from high-quality RNA, and smears indicated RNA degradation. The uncropped non-quantitative gel per *n* is shown; *18S* and *28S* peaks of sample #3 (Fresh) and #2 (Control, 5 days at 25 °C and 5 days at 40 °C). Note the different scales in Figure 1C. The dashed line indicates OD260/A280 = 2.0 for high-quality RNA. Effect of tube type in the linear regression models for each temperature point are indicated; *** *p* < 0.001, ** *p* < 0.01, * *p* < 0.05, ns, non-significant; RIN, RNA integrity number.

**Figure 2 ijms-23-10609-f002:**
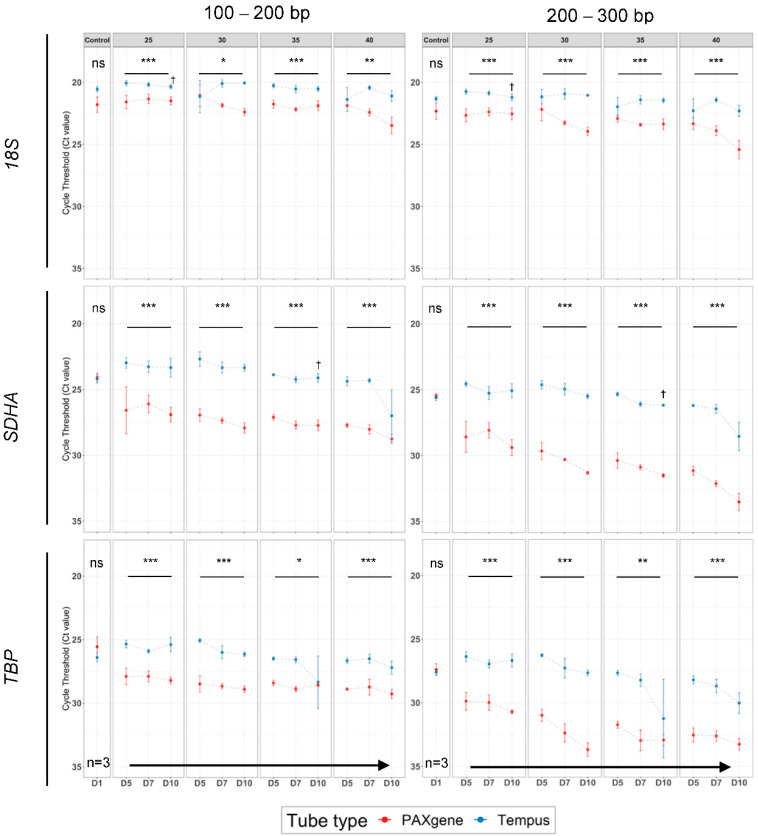
Cycle thresholds (Ct) of housekeeping genes assessing RNA from PAXgene^®^ and Tempus^™^ stored at different temperatures and times. The mean cycle threshold (Ct) values for *18S*, *SDHA,* and *TBP* with 100–200 base pair (bp) (short-amplicon) or 200–300 bp (medium-amplicon) lengths across different conditions (control and at multiple storage temperatures and days (matched *n* = 3)). *** *p* < 0.001, ** *p* < 0.01, * *p* < 0.05; ns, non-significant; †, data from only two observations were potentially available due to LOD. Blue, Tempus^™^ Blood RNA tubes; red, PAXgene^®^ Blood RNA tubes.

**Figure 3 ijms-23-10609-f003:**
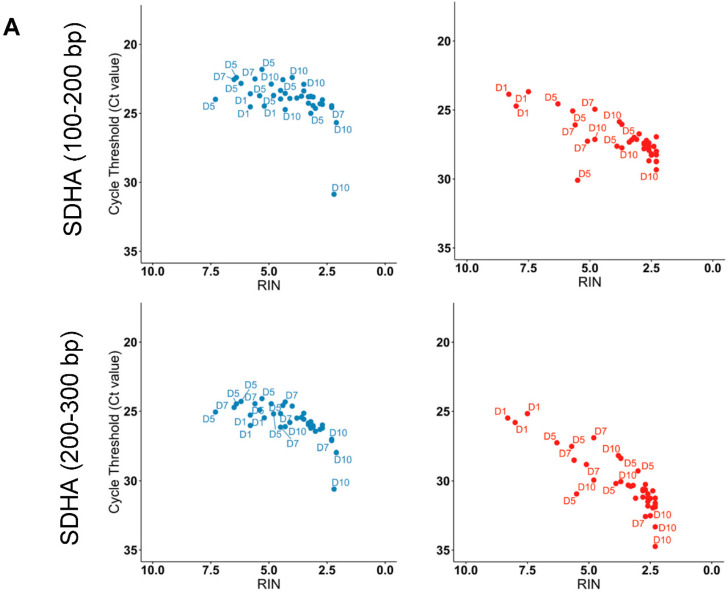
Cycle thresholds (Ct) in dependence on amplicon length and RNA integrity (RIN). (**A**) Scatter plots show the Ct values in dependence on amplicon length and RNA integrity (RIN) for different lengths of *SDHA* amplicons differences for each tube type. (**B**) Spearman correlation of the ΔCt values with RIN value (RNA quality) for medium-length amplicons (200–300 bp) of *18S* (**left**), *SDHA* (**middle**), and *TBP* (**right**). Blue, Tempus^™^ Blood RNA tubes; red, PAXgene^®^ Blood RNA tubes.

**Figure 4 ijms-23-10609-f004:**
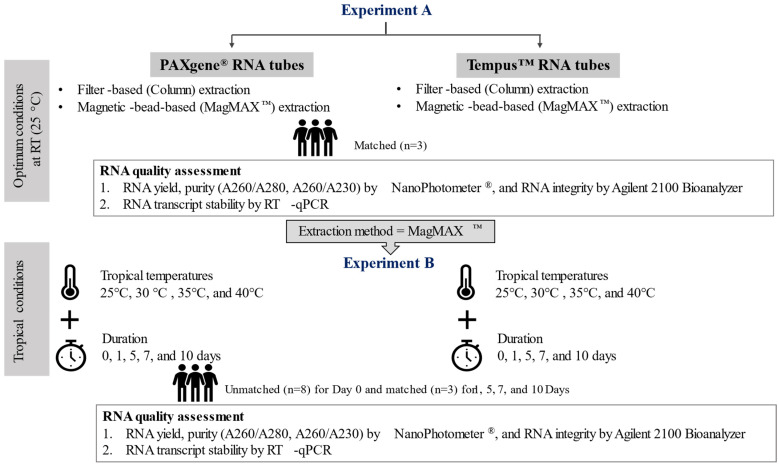
Experimental design. **Experiment A.** Comparison of total RNA yields, purity, and integrity using spin column-based and magnetic bead-based kits for the two types of blood stabilization systems (PAXgene^®^ Blood RNA Tubes and Tempus^™^ Blood RNA Tubes). All RNA samples were extracted fresh post collection. **Experiment B**. Systematic testing of different temperatures (25, 30, 35, or 40 °C) and storage times (0, 1, 5, 7, or 10 days) were immediately frozen at −80 °C for later extraction with matched samples (*n* = 3). Samples immediately frozen at −80 °C for later extraction is referred to “D1/Control”. Unmatched samples (*n* = 8) collected in PAXgene^®^ or Tempus^™^ RNA stabilizing systems and processed post collection immediately at optimal laboratory conditions (D0/Fresh).

**Table 1 ijms-23-10609-t001:** Multiple linear regression models. Model 1 on normalized RNA concentration (ng/μL), Model 2 on A260/A280 ratios, and Model 3 on RIN (expressed as log_2_ RIN).

**Model 1 = RNA Concentration (ng/μL)**				
Explanatory variable	Estimate	Std. Error	*t* value	*p*-value
(Intercept)	66.230	78.853	0.84	0.404
Tube type: Tempus^™^	254.894	47.339	5.384	1.15 × 10^−6^ ***
Days	5.049	9.997	0.505	0.615
Temperature	−0.106	2.358	−0.045	0.964
Tube-type Tempus^™^: Days	−5.859	3.342	−1.753	0.084
Tube-type Tempus^™^: Temperature	−4.650	1.228	−3.786	0.0003 *
Days: Temperature	−0.170	0.299	−0.57	0.571
Adjusted R-squared: 0.764 (*p* value: <2.2 × 10^−16^)				
**Model 2 = A260/A280**				
Explanatory variable	Estimate	Std. Error	t value	*p*-value
(Intercept)	2.181	0.102	21.466	<2 × 10^−16^ ***
Tube type: Tempus^™^	0.161	0.061	2.638	0.011 *
Days	0.008	0.013	0.606	0.547
Temperature	0.000	0.003	−0.085	0.933
Tube-type Tempus^™^: Days	−0.008	0.004	−1.951	0.055
Tube-type Tempus^™^: Temperature	−0.005	0.002	−3.075	0.003 **
Days: Temperature	0.000	0.000	−0.567	0.572
Adjusted R-squared:0.565 (*p*-value: 1.458 × 10^−10^)				
**Model 3 = log_2_(RIN)**				
Explanatory variable	Estimate	Std. Error	t value	*p*-value
(Intercept)	4.905	0.572	8.581	3.45 × 10^−12^ ***
Tube type: Tempus^™^	0.338	0.343	0.985	0.328
Days	−0.148	0.072	−2.047	0.045 *
Temperature	−0.082	0.017	−4.777	1.10 × 10^−5^ ***
Tube-type Tempus^™^: Days	−0.020	0.024	−0.836	0.407
Tube-type Tempus^™^: Temperature	0.002	0.009	0.236	0.814
Days: Temperature	0.002	0.002	1.057	0.295
Adjusted R-squared: 0.797 (*p* value: <2.2 × 10^−16^)				

Independent multiple linear regression models: Model 1 on normalized RNA concentration, Model 2 on A260/A280 ratios, and Model 3 on log_2_(RIN) values as dependent variable and tube types, days, temperatures, and subjects as the independent variables. *** *p* < 0.001, ** *p* < 0.01, * *p* < 0.05.

## Data Availability

The datasets generated during and/or analysed during the current study are available from the corresponding author on reasonable request.

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
