# Peer review of "The Effect of Tropical Temperatures on the Quality of RNA Extracted from Stabilized Whole-Blood Samples"

_ijms, 2022, doi:10.3390/ijms231810609_

Round 1

Reviewer 1 Report

In this manuscript, the authors analysed the RNA quality purified from whole blood samples by two collection systems, namely PAXgene and Tempus. The RNA sample qualities are critical for qPCR, RNAseq and other applications. The studies were performed in both optimal conditions and suboptimal conditions such as tropical conditions. In the latter conditions, the following parameters were tested, the duration of storage and high temperatures during storage. The results of the tests showed that Tempus maintained a higher quantity and quality of RNA than PAXgene in suboptimal tropical conditions. In terms of purity, both systems are equivalent. Altogether, the conducted experiments show that Tempus is a better solution than PAXgene for the preparation of RNA samples from a whole-blood collection in suboptimal conditions. The conducted experiments are properly exploited by careful statistic analysis. The conclusions of each experiment are well interpreted and bring interesting features about these two kits that will be useful for users.     

Specific point:

P 5, line 172-175, please replace ‘18s’ and ‘28s’ by ‘18S’ and ‘28S’. The same remark applies to figure 1C legend and throughout the whole manuscript. Moreover, the positions of 18S and 28S rRNA peaks and bands should be clearly indicated in figure 1C.  

Author Response

Specific point: P 5, line 172-175, please replace ‘18s’ and ‘28s’ by ‘18S’ and ‘28S’. The same remark applies to figure 1C legend and throughout the whole manuscript. Moreover, the positions of 18S and 28S rRNA peaks and bands should be clearly indicated in figure 1C

Response: We have revised the entire manuscript accordingly. Specifically, Figure 1C has been revised with positions of 18S and 28S rRNA peaks indicated on “Fresh” samples. In addition, Figure 2 has been updated with ‘18S’ in the figure legend.

Reviewer 2 Report

Authors reported results about the usage of two commercial blood collection systems for further RNA extraction/purification. In particular, they provide data about the quality of RNA extracted from blood comparing different temperatures and time of collection procedure. 

The work is well designed and well written.

Minor revision:

- Line 60-67: These sentences report results. I'd delete (or move to the abstract) them from the introduction.

- Line 213: please, specify some examples of the Ct value ... tableS5 is not so clear.

- Line 372: please, specify why you need to shake vigorously and how.

- Line: 474-477: you write about diagnosis etc ... related to NTDs. Do you mean using RNA as target ? biomarkers ? not all of NTDs' pathogens are analysed using RNA but DNA. 

Author Response

Minor revision: Line 60-67: These sentences report results. I'd delete (or move to the abstract) them from the introduction.

Response: Text has been revised accordingly. A couple of sentences have been moved to the Abstract and other text has been deleted from the introduction.

Minor revision: Line 213: please, specify some examples of the Ct value ... tableS5 is not so clear.

Response: Text has been revised as suggested, “For example, mean Ct values obtained for larger amplicons (200-300 bp) tested in PAXgene® tubes [22.32 (18S), 25.48 (SDHA), 27.39 (TBP)] and in Tempus tubes [21.33 (18S), 25.59 (SDHA), 27.59 (TBP)] were largely consistent.” Table S5 is provided as an excel file (.xlsx) format for clarity.

Minor revision: Line 372: please, specify why you need to shake vigorously and how.

Response: Text has been revised to specify these details, as follows: “immediately shaken vigorously for 10s to ensure that the stabilizing reagent makes uniform contact with the sample as per the manufacturer’s instructions.”

Minor revision: Line: 474-477: you write about diagnosis etc ... related to NTDs. Do you mean using RNA as target ? biomarkers ? not all of NTDs' pathogens are analysed using RNA but DNA

Response: Thank you for this helpful comment. We agree that not all NTD pathogens are RNA. We have revised the sentence as follows: “Our findings are especially relevant to research on RNA biology which could help future directions of diagnosis, treatment and interventions against diseases that are prevalent in tropical countries, including neglected tropical diseases.”